# Brain-related genes are specifically enriched with long phase 1 introns

**Eugene F. Baulin** [1,2]*, **Ivan V. Kulakovskiy** [1,3], **Mikhail A. Roytberg** [1,2†], **Tatiana V. Astakhova** [1]

**1** Institute of Mathematical Problems of Biology RAS—the Branch of Keldysh Institute of Applied Mathematics of Russian Academy of Sciences, Pushchino, Moscow Region, Russia, **2** Phystech School of Applied Mathematics and Informatics, Moscow Institute of Physics and Technology (National Research University), Dolgoprudny, Moscow Region, Russia, **3** Engelhardt Institute of Molecular Biology, Russian Academy of Sciences, Moscow, Russia

† Deceased.
* baulin@lpm.org.ru

## Abstract

Intronic gene regions are mostly considered in the scope of gene expression regulation, such as alternative splicing. However, relations between basic statistical properties of introns are much rarely studied in detail, despite vast available data. Particularly, little is known regarding the relationship between the intron length and the intron phase. Intron phase distribution is significantly different at different intron length thresholds. In this study, we performed GO enrichment analysis of gene sets with a particular intron phase at varying intron length thresholds using a list of 13823 orthologous human-mouse gene pairs. We found a specific group of 153 genes with phase 1 introns longer than 50 kilobases that were specifically expressed in brain, functionally related to synaptic signaling, and strongly associated with schizophrenia and other mental disorders. We propose that the prevalence of long phase 1 introns arises from the presence of the signal peptide sequence and is connected with 1–1 exon shuffling.

**Data Availability Statement:** All relevant data are within the paper and its Supporting Information files.

**Funding:** The authors received no specific funding for this work.

## Introduction

The exon-intron structure of genes is essential for cell type- and condition-specific gene expression regulation [1], maintaining genetic stability [2] and genome evolution [3–6] of higher eukaryotes.

Comprehensive genome annotation allows studying the basic statistical properties of introns. Particularly, introns are usually characterized by an ordinal number within the gene, the region length, and features of the nucleotide sequence. Another important numeric characteristic is the phase, i.e. the sum of the lengths of the preceding exons modulo 3, e.g. the phase 1 intron occurs after the first nucleotide of a codon.

It's known that there is an excess of phase 0 introns in all eukaryotic genomes [7]. The usual ratio of intron phases 0, 1, and 2 is nearly 5:3:2 [7]. The phase distribution of spliceosomal introns strongly correlates with the sequence conservation of splicing signals in the

**Competing interests:** The authors have declared that no competing interests exist.

neighboring exons [8]. The relatively underrepresented phase 2 introns exhibit the lowest conservation rate, the relatively overrepresented phase 0 introns show the highest conservation level, and phase 1 introns have no special properties in regard to sequence conservation. In [9] it was shown that the creation of new exon-intron structures through genetic recombination, the exon shuffling, in metazoans is predominantly mediated by phase 1 introns (1–1 exon shuffling), whereas 0–0 exon shuffling prevails in non-metazoan organisms. 2–2 exon shuffling rate was insignificant in all organisms. Based on these data, the authors hypothesized that phase 0 introns are more ancient than phase 1 and 2 introns, but claimed that the genetic and selection mechanisms that lie behind the preference for shuffling 1–1 exons or domains remain largely unknown.

In [10] authors found a significant excess of phase 1 introns in the vicinity of the signal peptide cleavage site in human genes. It was suggested that the amino acid sequence of the signal peptide favors an enrichment of phase 1 AG|G proto-splice sites in the vicinity of the cleavage site and the depletion of these sites in any phase within the peptide-encoding RNA sequence. However, in [11] authors claimed there were no disproportional excess of phase 1 AG|G sites in the vicinity of the cleavage site and suggested an alternative scenario, where the excess of phase 1 introns was produced by 1–1 exon shuffling

A study of the *H. sapiens*, *D. melanogaster*, *C. elegans*, and *A. thaliana* genomes [12] showed increasing frequency of phase 0 introns and decreasing frequency of phase 1 when scanning genes from 5' to 3'. This tendency was specifically exhibited in the genomes of *Homo sapiens* and *Arabidopsis thaliana*, for which the emergence of new introns in the 3' region of genes was suggested to be the dominant process [12]. The authors also consider the high frequency of phase 1 introns immediately after the signal peptide to be a minor contributing factor for the observed gradient.

In [13] it was demonstrated that introns in tissue-specific and development-specific genes are longer than those in housekeeping genes. It was claimed that the share of the conserved sequence is higher in tissue-specific genes, which may indicate that the intron evolution is not neutral in general. It was also reported that the length of the conserved intronic DNA in a gene was correlated with the number of functional domains in the protein encoded by that gene. According to [14], the intronic burden of a gene (considering either the total length or the number of introns) is positively correlated with its evolutionary conservation which suggests functional importance of the genes with large intronic regions.

In [15] the authors claimed that the first introns are usually longer in eukaryotic genes, for first introns both inside 5'-untranslated regions (5' UTRs) or coding sequences (CDSs). A proposed explanation was that the increased length of first introns allowed presence of additional functional elements compared to 'normal' intron structure. Consistently, the fraction of conserved sequence in the first introns was found higher compared to that of all the downstream introns (see Fig 3A in [13]).

Despite the vast amount of information that is already known about introns, there is still a lack of work regarding the relationship between the intron length and the intron phase.

In our previous work [16] we considered 17 animal genomes of various taxons (including insects, fish, amphibians, reptiles, birds, and mammals) and demonstrated that shares of intron phases significantly varied with increasing lengths of considered introns, particularly, the share of phase 1 increased and reached phase 0 share at a certain threshold (~100 kilobases (kb) in mammals).

In this work, we performed functional annotation of human and mouse orthologs satisfying particular intron phase and length constraints.

## Materials and methods

We used GENCODE [17] gene annotations for *Homo sapiens* hg38 (GENCODE version 28) and *Mus musculus* mm10 (GENCODE version M18) genomes. The set of considered genes has been limited to protein-coding curated (HAVANA) transcripts with the transcript support level of 1. For each gene with several eligible transcripts, we selected the one with the maximum number of exons. Human-mouse orthologous pairs were extracted from the MGI database [18]. The resulting set consisted of 13823 orthologous gene pairs.

All introns were segregated into 4 groups by length: less than 5 kb (denoted <5 below), between 5 kb and 10 kb (5–10), between 10 kb and 50 kb (10–50), and longer than 50 kb (>50). Intron phases were marked ph0, ph1, and ph2 respectively for phase 0, phase 1, and phase 2. Every gene from orthologous pairs was annotated with the number of introns belonging to each of 12 possible 'phase-length' groups, see S1 Table.

For GO enrichment analysis, the list of gene ontology (GO) terms associated with each gene was extracted from MGI [18]. Each pair of a 'phase-length' group of introns and a GO-term was annotated with the number of orthologs associated with the GO-term and containing at least one intron of the given group (a) both in human and mouse, (b) in human only, (c) in mouse only, and (d) not containing such introns neither in human nor mouse, see S2 Table. With these data, we used Fisher's exact test [19] to estimate the statistical significance of the GO term enrichment for each 'phase-length' group. P-values were corrected for multiple tested pairs using Benjamini-Hochberg (FDR) procedure [20]. GO terms with significant enrichment (passing 1% FDR) in both human and mouse were selected for further analysis, see S3 Table.

## Results

Phase 1 introns demonstrate a distinct pattern with multiple GO terms enriched in the group of length > 50 kb unlike phases 0 and 2 that demonstrate strong enrichments of multiple GO terms only for shorter introns (Fig 1).

To perform an in-depth analysis, we created a subset of genes with ph1_>50 introns by gathering all such genes with at least one GO term enriched in ph1_>50 introns only (see Table 1 for the list of GO-terms). The resulting list consisted of 153 genes composed of 2235 exons, see S4 Table. Within CDSs, the genes included 742, 917, and 423 introns in phases 0, 1, and 2 respectively. Consistently, the selected genes showed a significant abundance of phase 1 introns longer than 50 kilobases (S1 Fig), while in the complete set of genes shares of phases 0 and 1 introns longer than 50 kilobases were comparable.

Assuming the Gene Ontology annotation is not comprehensive, we assembled the complete list of genes with ph1_>50-introns (507 genes) for further analysis, see S4 Table. Indeed, analysis of tissue-specific gene expression showed that both 507 and 153 groups are specifically expressed in brain tissues (S2 Fig).

We also performed an additional enrichment analysis of the 153 genes with respect to associated diseases having the entire set of 13823 genes as background. In agreement with brain-specific expression, we found associations with a number of mental disorders (S3 Fig), including schizophrenia and depression, which were detected as significant in three out of four annotation databases.

The excess of long phase 1 introns in the primary sublist of 153 genes can be caused by the presence of the signal peptide or with the 1–1 exon shuffling events.

Indeed, 83 out of 153 (54%) genes started with the signal peptide sequence (according to the UniprotKB [21], see S4 Table), which is significantly greater than the overall percentage of 17% in human genes (3596 out of 20365, according to the UniprotKB [21]). Furthermore, long phase 1 introns are preferably found at the 5'-end of the spliced transcripts (Fig 2). Next, from

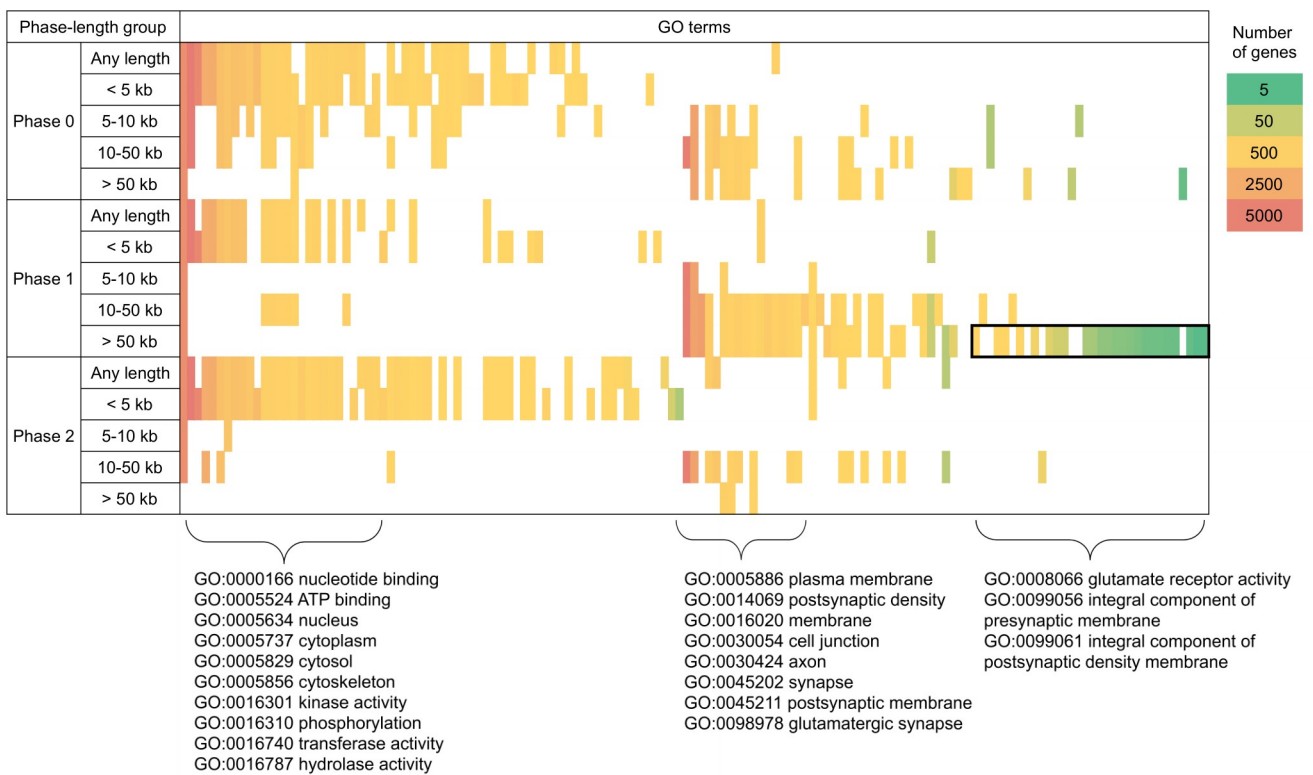

**Fig 1. Genes containing introns of particular length and phase are enriched with multiple GO-terms.** Rows—'phase-length' intron groups, columns—GO-terms. The cells with significant enrichment (1% FDR) are colored. Color scale reflects the total number of genes associated with a particular GO term. Columns have been arranged manually by placing the GO-terms enriched with the shorter groups of introns on the left, GO-terms enriched with the groups of long introns on the right, and then sorting within those two halves of GO-terms by their sizes keeping similar rows close to each other. The GO-terms enriched with phase 1 introns longer than 50 kilobases only are emphasized with the black frame. Curly brackets mark out the clusters of similar GO-terms and are followed with the particular terms that demonstrate the most significant enrichments.

9 possible phase types of exons, only 1–1 exons are significantly enriched in the 153 genes (odds ratio of 2, FDR-corrected P-value $< 10^{-30}$, see S4 Table). Thus, the observed enrichment of long phase 1 introns is associated with both the presence of the signal peptide and with the 1–1 exon shuffling.

## Discussion

As it's mentioned in Introduction, according to [11] the connection between phase 1 introns and the signal peptide may be explained by 1–1 exon shuffling events, and, according to [9], the preference for shuffling 1–1 exons remain largely unknown. We propose the following scenario to explain the observed associations.

First, the prevalence of phase 1 introns immediately after the signal peptide sequence [10] is caused by the conserved A, V, and G amino acids in the vicinity of the cleavage site [22]. This conservation is dictated by the (-3, -1) rule that states the residues at positions -3 and -1 (relative to the cleavage site) must be small and neutral for correct cleavage [22]. These three amino acids are coded by GNN codons which favor phase 1 introns since among the three positions of the AG|G common exon-exon junction motif only the G| position is strongly conserved and the other two are hardly specific [8]. The prevalence of phase 0 introns immediately after the ATG start codons (i.e. between codons 1 and 2 of the gene) observed in [23] can serve as

**Table 1. 25 GO-terms significantly associated with genes of the ph1_>50 introns group.**

| GO-term | Number of associated genes with ph1>50 introns | Total number of associated genes | Odds ratio | FDR |
|---|---|---|---|---|
| GO:0007165 signal transduction | 63 | 739 | 2.68 | 5.00E-06 |
| GO:0099056 integral component of presynaptic membrane | 21 | 77 | 10.32 | 5.08E-09 |
| GO:0050804 modulation of chemical synaptic transmission | 18 | 77 | 8.34 | 4.26E-06 |
| GO:0031225 anchored component of membrane | 17 | 110 | 4.97 | 8.9E-03 |
| GO:0099061 integral component of postsynaptic density membrane | 17 | 54 | 12.56 | 7.35E-08 |
| GO:0099055 integral component of postsynaptic membrane | 16 | 67 | 8.55 | 3.07E-05 |
| GO:0043195 terminal bouton | 15 | 72 | 7.15 | 7.2E-04 |
| GO:0051965 positive regulation of synapse assembly | 13 | 60 | 7.49 | 3.6E-03 |
| GO:2000300 regulation of synaptic vesicle exocytosis | 13 | 54 | 8.59 | 9.4E-04 |
| GO:0007157 heterophilic cell-cell adhesion via plasma membrane cell adhesion molecules | 12 | 38 | 12.50 | 1.1E-04 |
| GO:0007416 synapse assembly | 12 | 37 | 13.00 | 7.57E-05 |
| GO:0051966 regulation of synaptic transmission, glutamatergic | 11 | 25 | 21.25 | 7.36E-06 |
| GO:0032281 AMPA glutamate receptor complex | 10 | 25 | 17.99 | 1.4E-04 |
| GO:0008066 glutamate receptor activity | 10 | 11 | 270.16 | 7.81E-10 |
| GO:0005246 calcium channel regulator activity | 9 | 28 | 12.75 | 7.4E-03 |
| GO:0035249 synaptic transmission, glutamatergic | 9 | 28 | 12.75 | 7.4E-03 |
| GO:0051968 positive regulation of synaptic transmission, glutamatergic | 9 | 23 | 17.32 | 1.0E-03 |
| GO:0004970 ionotropic glutamate receptor activity | 9 | 15 | 40.43 | 8.35E-06 |
| GO:0046328 regulation of JNK cascade | 8 | 20 | 17.92 | 4.7E-03 |
| GO:0036477 somatodendritic compartment | 8 | 16 | 26.89 | 5.5E-04 |
| GO:2000311 regulation of AMPA receptor activity | 7 | 15 | 23.48 | 7.6E-03 |
| GO:0005003 ephrin receptor activity | 7 | 14 | 26.84 | 4.2E-03 |
| GO:0005005 transmembrane-ephrin receptor activity | 7 | 14 | 26.84 | 4.2E-03 |
| GO:0015277 kainate selective glutamate receptor activity | 5 | 6 | 133.72 | 6.8E-03 |
| GO:0032983 kainate selective glutamate receptor complex | 5 | 6 | 133.72 | 6.8E-03 |

The first column lists GO-term accession numbers and their descriptions. The last column shows the FDR-corrected significance of the corresponding enrichment.

the evidence that a single G| position is enough to favor a specific intron phase. In turn, sharp depletion of introns in the middle of the signal peptide coding region [10] may be explained by the fact that the region can serve as a signal for an alternative nuclear export of an mRNA [24].

The excess of long phase 1 introns (and not short introns) is possibly caused by the generally longer first introns in eukaryotes [15]. The signal peptide sequence is always on the 5'-end of the gene and consequently the intron immediately after it is often the first intron of the gene.

Next, we state the preference for shuffling 1–1 exons is caused exactly by the excess of phase 1 introns immediately after the signal peptide sequence (and not in the opposite direction of causality as earlier proposed in [11]).

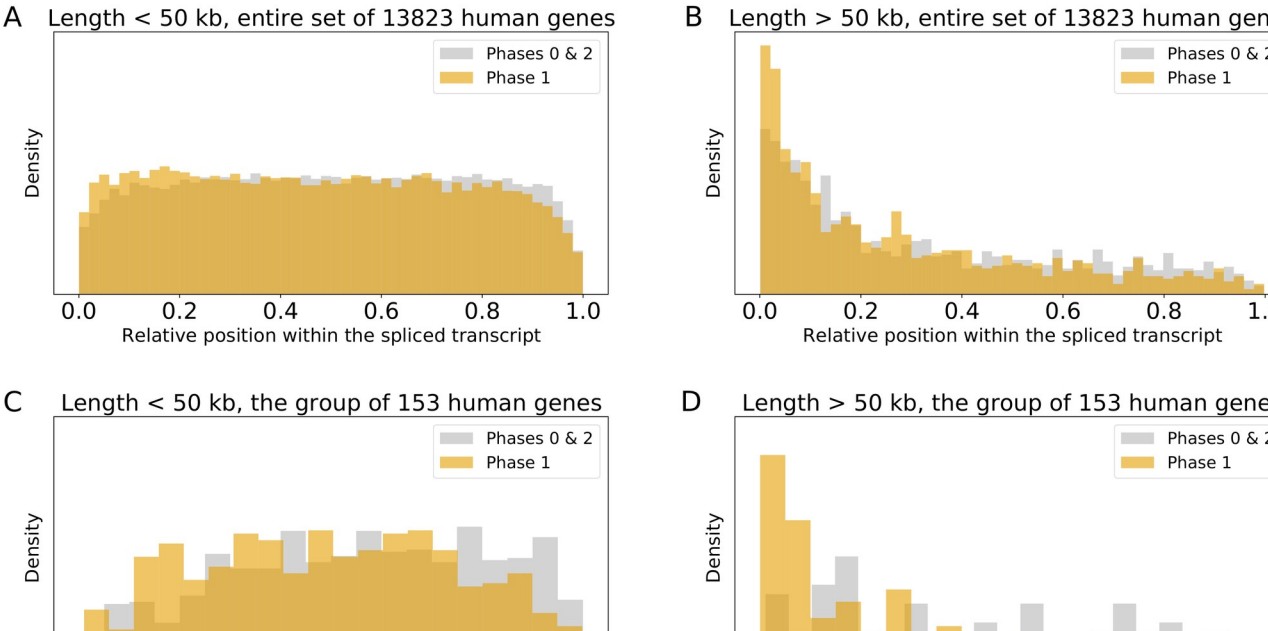

**Fig 2. Distribution of relative intron positions along the transcripts.** (A) Introns shorter than 50 kilobases of the entire set of considered human genes. (B) Introns longer than 50 kilobases of the entire set of considered human genes. (C) Introns shorter than 50 kilobases of the group of 153 genes. (D) Introns longer than 50 kilobases of the group of 153 genes. Y axis: distribution density, X axis: the relative position of an intron within the spliced transcript. Phase 1 introns are shown in orange, phase 2 and 0 introns are shown together in gray.

First, there are several examples of the evolution governed by exon shuffling events in the genes composed of the signal peptide sequence followed by a phase 1 intron and another protein domain coding region, in particular, this was observed for serine proteases [25, 26], β-defensins [27], and scorpion venom genes [28].

Second, according to [29], the extracellular proteins in mammals evolve faster than the intracellular ones, and extracellular protein must start with the signal peptide. The observed faster evolution must have been governed by exon shuffling events since the modular proteins are mainly membrane and secreted proteins and the modular domains are mainly coded by 1–1 exons [3, 4, 30, 31]. The additional argument in favor of the above statement is that most of the recently duplicated genes encode membrane and secreted proteins [32].

We believe that such fast evolution caused by 1–1 exon shuffling close to the 5'-end of the genes should have been made a significant contribution to the tendency of phase 1 to decrease and phase 0 to increase in the direction from 5'- to 3'-end of the gene observed in [12] (see Fig 2).

The selected group of 153 genes with a great excess of long phase 1 introns is consistently enriched with brain-specific genes, which are considered evolutionarily young and enriched with alternative splicing events [33], requiring high intronic burden.

## Supporting information

**S1 Fig. Phase distribution of long (longer than 50 kilobases) introns.**
(PNG)

**S2 Fig. Tissue-specific gene enrichment.** (a) All 507 genes containing ph1>50 introns. (b) 153 genes selected based on GO enrichment. Graphs have been generated with TissueEnrich [1]. (c) The top 50 genes (out of the 153 genes) with the largest total phase 1 introns size. The graph is generated with GTEx Multi Gene Query tool [2].
(PNG)

**S3 Fig. Disease and pathway enrichment analysis of the 153 genes.** Graphs have been generated with WebGestalt [3].
(PNG)

**S1 Table. Number of introns of different phase and length in human-mouse orthologous gene pairs.**
(XLSX)

**S2 Table. Number of human-mouse orthologous gene pairs containing introns of different phase and length associated with particular GO-terms.**
(XLSX)

**S3 Table. GO-enrichment of specific phase-length intron groups.**
(XLSX)

**S4 Table. Lists of human gene names of 153 orthologous human-mouse gene pairs of interest and of 507 orthologous human-mouse gene pairs containing ph1>50 introns both in human and mouse.** Particular spreadsheets present exon-intron structure of the 153 genes, the first phase 1 introns, and the unique functional protein domains coded by the 1–1 exons of the 153 genes. 1. Jain A, Tuteja G. TissueEnrich: Tissue-specific gene enrichment analysis. *Bioinformatics*. 2019 Jun 1;35(11):1966–7. doi: 10.1093/bioinformatics/bty890 2. Carithers LJ, Moore HM. The Genotype-Tissue Expression (GTEx) Project. *Biopreservation and Biobanking*. 2015 Oct 1;13(5):307. doi: 10.1089/bio.2015.29031.hmm 3. Wang J, Vasaikar S, Shi Z, Greer M, Zhang B. WebGestalt 2017: a more comprehensive, powerful, flexible and interactive gene set enrichment analysis toolkit. *Nucleic acids research*. 2017 Jul 3;45(W1):W130-7. doi: 10.1093/nar/gkx356.
(XLSX)

## Author Contributions

**Conceptualization:** Mikhail A. Roytberg, Tatiana V. Astakhova.

**Data curation:** Eugene F. Baulin, Tatiana V. Astakhova.

**Investigation:** Eugene F. Baulin, Ivan V. Kulakovskiy, Tatiana V. Astakhova.

**Methodology:** Eugene F. Baulin, Ivan V. Kulakovskiy, Mikhail A. Roytberg, Tatiana V. Astakhova.

**Project administration:** Tatiana V. Astakhova.

**Supervision:** Ivan V. Kulakovskiy.

**Validation:** Ivan V. Kulakovskiy.

**Visualization:** Eugene F. Baulin.

**Writing – original draft:** Eugene F. Baulin.

**Writing – review & editing:** Ivan V. Kulakovskiy, Tatiana V. Astakhova.

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
