## [Decision Letter · Decision Letter 0]

6 Apr 2020

PONE-D-20-04197

Brain-related genes are specifically enriched with long phase 1 introns

PLOS ONE

Dear Mr. Baulin,

Thank you for submitting your manuscript to PLOS ONE. After careful consideration, we feel that it has merit but does not fully meet PLOS ONE’s publication criteria as it currently stands. Therefore, we invite you to submit a revised version of the manuscript that addresses the points raised during the review process.

The paper was reviewered by 2 experts in the field. They have several suggestions, please, try to implement them. I think that some suggested experiments may be hard to implement. For example, the alternative splicing is an interesting topic but the amount of isoforms is huge for some genes. I am not sure how to make a "the background random model", may be you will have an idea.

We would appreciate receiving your revised manuscript by May 21 2020 11:59PM. To enhance the reproducibility of your results, we recommend that if applicable you deposit your laboratory protocols in protocols.io, where a protocol can be assigned its own identifier (DOI) such that it can be cited independently in the future. For instructions see: http://journals.plos.org/plosone/s/submission-guidelines#loc-laboratory-protocols

We look forward to receiving your revised manuscript.

Kind regards,

Igor B. Rogozin

Academic Editor

PLOS ONE

Additional Editor Comments (if provided):

Dear Dr. Baulin,

The paper was reviewered by 2 experts in the field. They have several suggestions, please, try to implement them. Some experiments may be hard to implement. For example, the alternative splicing is an interesting topic but the amount of isoforms is huge for some genes. I am not sure how to make a "the background random model", may be you will have an idea.

Best regards, Igor

Reviewers' comments:

Reviewer's Responses to Questions

**Comments to the Author**

1. Is the manuscript technically sound, and do the data support the conclusions?

Reviewer #1: Partly

Reviewer #2: Yes

2. Has the statistical analysis been performed appropriately and rigorously? 

Reviewer #1: No

Reviewer #2: Yes

3. Have the authors made all data underlying the findings in their manuscript fully available?

Reviewer #1: Yes

Reviewer #2: Yes

4. Is the manuscript presented in an intelligible fashion and written in standard English?

Reviewer #1: Yes

Reviewer #2: Yes

5. Review Comments to the Author

Reviewer #1: Review on the manuscript titled “Brain-related genes are specifically enriched with long phase 1 introns” by Baulin et al., 2020

The authors addressed the intron phase distribution dependence on length in 14000 human-mouse orthologous pairs. They provide phase distribution data in supplementary for mouse and human in 5 intron length bins:

0<5kb<10kb<50kb<~

Next the authors checked each bin specific genes (intron-wise) for GO abundance, result provided in supplementary. They identified 153 genes (and how many introns?) in >50kb bin were abundant in phase 1 (suppl fig1), and the genes were brain specific (suppl fig2). The authors noted specific codon distribution in the target sample.

While the issue seems interesting, there are multiple things that preclude publication of the draft in the current form. They are:

1) The authors claim that the major biological impact of phase skew is codon frequency bias. While there could be some skewed codon preference, I personally doubt that it could be the major trait associated with the phenomena. There are some previous articles on the phase 1 bias in signal peptide – containing genes (and it is the sample of more than 2000 genes), but no definite conclusions where made ever since the phenomenon’s been reported. Also, this fact should be highlighted in the introduction and discussion.

2) The authors made the major stress on GO ontology relation to the particular bins and phases: both tables S2 and S3 are devoted to that. It’s my particular point that GO enrichment effect is rather speculative and cannot be (was not in the manuscript) the basis for any definite inference unless explicitly proved by some distinct biological hypothesis testing, or at least stating. Pooling the genes intron-wise look rather morbid for reader screening the data as well as overall sense exploring it (problem statement issue). There is 'viewable' and 'accessible' results in 153 ‘more than 50kb’ bin onl , but it is the subject of more profound elaboration as I mentioned above

3) There’s no obvious biological ground/model of manifested relations provided in manuscript.

4) The authors provided multiple comparison scheme in the course of GO enrichment analysis Tables S2, S3), but no correction for multiple comparisons is observed (FDR)

5) The figures and tables in the manuscript are quite lack of explicit description of the fields by the legends, while they are assumed to be self-sufficient (Table 2). Why there’s a log score field, but no significance value? It would be good to mention the number of instances observed for each pattern in Table 2. Fig. 1 is quite awkward. The names are turned bottom to the top and overall non-transparent, it is advised to redesign it. Cannot see Fig 1 Title. Maybe the authors would think of redesigning tables to phase vs bin projection, so they would be 2-dimensional, for the sake of more apprehensive view. I think the authors already speculated on this issue, so not completely sure on that. No description and column titles in Supplem. Table 3, last spreadsheet.

6) In the text: S1 Fig -> Fig. 1; S6 Table-> Table S6, etc.

7) 153 genes are not specifically cortex genes, but rather brain-specific ones. Most of them express in various brain regions. So they may be called as brain-specific genes, but I'm not insisting on that.

8) It is known that brain- specific genes are quite long and are enriched for alternative splicing rates. I haven’t found any correspondent brain specific genes features description in the manuscript. I suggest to check the intron phases locations distribution in the set as well as to assess overall AS enrichment in these genes, their length and exons number distribution. I assessed median # exons=15, median length=351kb, and 2.6 isoforms per gene on average in 153 genes sample, which is significantly larger than genome average. Given large exons number per gene, the phase 1 exons would have a distinct elevated RANDOM chance to occur within the large introns. In particular, I found that there is a definite (significant) overall skew of phase 1 introns in the 153-fold sample: 898: 911: 423 (0,1,2 phases). At least 46 genes within a sample contain signal peptide at the beginning, though it may not greatly impact the results. So, it may be relevant to assess the background random model for this particular sample.

Other than that, since 153 genes can affect at most 200 codons on the phase 1 splice junction sites which is rather small amount, and given the previous failed attempts to assign codon bias as a target trait for the phase skew phenomenon in signal peptide motif containing genes, I observe the lack of the basic mechanistic hypothesis for the phenomenon in the manuscript, though it does look intriguing at this stage.

Reviewer #2: The authors found that "the observed increased share of phase 1 introns is linked with specific codon usage of brain-related genes, but further analysis is necessary to fully interpret of intron phase-length dependencies." I do not think that this is the best conclusion for a paper (it is the last sentence of the Abstract). "Further analysis" sounds kind of provocative. I wonder if the authors can provide something more informative as a conclusion in the Abstract.

1) The issue of codon usage of brain-related genes was discussed in details in Genome Biol Evol. 2018 Aug 1;10(8):1902-1919. doi: 10.1093/gbe/evy146.

Genome-Wide Changes in Protein Translation Efficiency Are Associated with Autism.

Rogozin IB1, Gertz EM1, Baranov PV2, Poliakov E3, Schaffer AA1.

The brain-specific codon usage is a controversial topic, it is better to desrcibe it more detail.

2) The observed phase tendencies may be a result of phase 1-1 domain shuffling, there is a discussion of this issue in

Domain mobility in proteins: functional and evolutionary implications.

Basu MK, Poliakov E, Rogozin IB.

Brief Bioinform. 2009 May;10(3):205-16.

Minor revision:

The Table 1 looks really bad, the names in the first column are truncated. May be formatting issues.

6. PLOS authors have the option to publish the peer review history of their article (what does this mean?). If published, this will include your full peer review and any attached files.

Reviewer #1: No

Reviewer #2: No

---

## [Author Response · Author response to Decision Letter 0]

9 May 2020

Additional Editor Comments (if provided):

Dear Dr. Baulin,

The paper was reviewered by 2 experts in the field. They have several suggestions, please, try to implement them. Some experiments may be hard to implement. For example, the alternative splicing is an interesting topic but the amount of isoforms is huge for some genes. I am not sure how to make a "the background random model", may be you will have an idea.

Best regards, Igor

=== 

Dear Prof. Rogozin,

We wholeheartedly thank you and the reviewers for careful evaluation of our manuscript and very valuable comments and suggestions. We did our best to revise the manuscript accordingly. Particularly, we extended Results, and fully rewrote Discussion. Please find our point-by-point response below.

On behalf of all authors,

Eugene Baulin

----

Reviewer #1: Review on the manuscript titled “Brain-related genes are specifically enriched with long phase 1 introns” by Baulin et al., 2020

The authors addressed the intron phase distribution dependence on length in 14000 human-mouse orthologous pairs. They provide phase distribution data in supplementary for mouse and human in 5 intron length bins: 0<5kb<10kb<50kb<~

Next the authors checked each bin specific genes (intron-wise) for GO abundance, result provided in supplementary. They identified 153 genes (and how many introns?) in >50kb bin were abundant in phase 1 (suppl fig1), and the genes were brain specific (suppl fig2). The authors noted specific codon distribution in the target sample.

===

We added the missing numbers (exons and introns) to the corresponding paragraph in the Results section.

----

1) The authors claim that the major biological impact of phase skew is codon frequency bias. While there could be some skewed codon preference, I personally doubt that it could be the major trait associated with the phenomena. There are some previous articles on the phase 1 bias in signal peptide – containing genes (and it is the sample of more than 2000 genes), but no definite conclusions where made ever since the phenomenon’s been reported. Also, this fact should be highlighted in the introduction and discussion.

===

We specifically thank the reviewer for this comment. We performed an additional analysis and realized the bias to be indeed related to the presence of the signal peptides. We have updated the Results and Discussion accordingly, and included the relevant references in the Introduction.

----

2) The authors made the major stress on GO ontology relation to the particular bins and phases: both tables S2 and S3 are devoted to that. It’s my particular point that GO enrichment effect is rather speculative and cannot be (was not in the manuscript) the basis for any definite inference unless explicitly proved by some distinct biological hypothesis testing, or at least stating. Pooling the genes intron-wise look rather morbid for reader screening the data as well as overall sense exploring it (problem statement issue). There is 'viewable' and 'accessible' results in 153 ‘more than 50kb’ bin onl , but it is the subject of more profound elaboration as I mentioned above

===

We have updated the Supplementary table 4 to include detailed information regarding the exon-intron structure of the selected genes as well as predicted Pfam domains coded by 1-1 exons, transmembrane domains and signal peptides. 

----

3) There’s no obvious biological ground/model of manifested relations provided in manuscript.

===

We did our best to re-check the related literature, and (including analysis related to comment 1) were able to formulate the hypothesis, explaining the observed phenomenon. We have updated the Discussion accordingly.

----

4) The authors provided multiple comparison scheme in the course of GO enrichment analysis Tables S2, S3), but no correction for multiple comparisons is observed (FDR)

===

The presented numbers were in fact FDR corrected for multiple tested pairs of a GO-term and a phase-length intron group. We changed the corresponding titles in the Supplementary table 3.

----

5) The figures and tables in the manuscript are quite lack of explicit description of the fields by the legends, while they are assumed to be self-sufficient (Table 2). Why there’s a log score field, but no significance value? It would be good to mention the number of instances observed for each pattern in Table 2. 

===

We have removed Table 2 in the revised version of the manuscript. We have added the missing caption to Table 1.

----

Fig. 1 is quite awkward. The names are turned bottom to the top and overall non-transparent, it is advised to redesign it. Cannot see Fig 1 Title. 

===

We have significantly redesigned Figure 1 to improve visual clarity. 

----

No description and column titles in Supplem. Table 3, last spreadsheet.

===

We have updated the Supplementary Table 3 which now includes a missing readme section with detailed description of the columns.

----

6) In the text: S1 Fig -> Fig. 1; S6 Table-> Table S6, etc.

===

In the original submission we followed the PLoS one guidelines regarding naming of supplementary figures and tables ("S1 Fig", "S1 Table" - as suggested in https://journals.plos.org/plosone/s/file?id=wjVg/PLOSOne_formatting_sample_main_body.pdf). In case the used naming scheme is incorrect, we believe it can be resolved at the technical editing stage.

----

7) 153 genes are not specifically cortex genes, but rather brain-specific ones. Most of them express in various brain regions. So they may be called as brain-specific genes, but I'm not insisting on that.

===

Fixed as suggested.

----

8) It is known that brain- specific genes are quite long and are enriched for alternative splicing rates. I haven’t found any correspondent brain specific genes features description in the manuscript. I suggest to check the intron phases locations distribution in the set as well as to assess overall AS enrichment in these genes, their length and exons number distribution. I assessed median # exons=15, median length=351kb, and 2.6 isoforms per gene on average in 153 genes sample, which is significantly larger than genome average. Given large exons number per gene, the phase 1 exons would have a distinct elevated RANDOM chance to occur within the large introns. In particular, I found that there is a definite (significant) overall skew of phase 1 introns in the 153-fold sample: 898: 911: 423 (0,1,2 phases). At least 46 genes within a sample contain signal peptide at the beginning, though it may not greatly impact the results. So, it may be relevant to assess the background random model for this particular sample.

===

Indeed, 71 genes out of 153 carry a (predicted) signal peptide at 5' end. We were able to link this observation with phase 1 introns, and updated the Discussion accordingly. The reference to a review describing brain-specific features of the alternative splicing is included in the second-last paragraph of Discussion.

----

Other than that, since 153 genes can affect at most 200 codons on the phase 1 splice junction sites which is rather small amount, and given the previous failed attempts to assign codon bias as a target trait for the phase skew phenomenon in signal peptide motif containing genes, I observe the lack of the basic mechanistic hypothesis for the phenomenon in the manuscript, though it does look intriguing at this stage.

===

This suggestion allowed us to significantly revise the manuscript. In the revised version, we argue that the codon bias in the vicinity of the cleavage site of the signal peptide is the reason for the intron phase skew. We explain the previous attempts to connect the phase skew with codon bias were unsuccessful due to the limited sequence conservation of the AG|G exon-exon junction motif since its two flanking positions are very weak compared to the major G| position.

----

Reviewer #2: The authors found that "the observed increased share of phase 1 introns is linked with specific codon usage of brain-related genes, but further analysis is necessary to fully interpret of intron phase-length dependencies." I do not think that this is the best conclusion for a paper (it is the last sentence of the Abstract). "Further analysis" sounds kind of provocative. I wonder if the authors can provide something more informative as a conclusion in the Abstract.

===

We fully agree with this comment. We have revised the Abstract as well as a large part of the Introduction and the whole Discussion section.

----

1) The issue of codon usage of brain-related genes was discussed in details in Genome Biol Evol. 2018 Aug 1;10(8):1902-1919. doi: 10.1093/gbe/evy146.

Genome-Wide Changes in Protein Translation Efficiency Are Associated with Autism.

Rogozin IB1, Gertz EM1, Baranov PV2, Poliakov E3, Schaffer AA1.

The brain-specific codon usage is a controversial topic, it is better to desrcibe it more detail.

===

We were not able to provide stronger evidence supporting the significant involvement of the codon usage, and this part of our analysis was also criticized by the 1st reviewer. We have removed this subsection from the revised manuscript.

----

2) The observed phase tendencies may be a result of phase 1-1 domain shuffling, there is a discussion of this issue in

Domain mobility in proteins: functional and evolutionary implications.

Basu MK, Poliakov E, Rogozin IB.

Brief Bioinform. 2009 May;10(3):205-16.

===

Indeed, the observed prevalence of phase 1 introns was found to be connected with 1-1 exon shuffling coupled with the presence of the signal peptide.

----

Minor revision:

The Table 1 looks really bad, the names in the first column are truncated. May be formatting issues.

===

FIXED.

----

---

## [Editor Report · Decision Letter 1]

18 May 2020

Brain-related genes are specifically enriched with long phase 1 introns

PONE-D-20-04197R1

Dear Dr. Baulin,

We are pleased to inform you that your manuscript has been judged scientifically suitable for publication and will be formally accepted for publication once it complies with all outstanding technical requirements.

With kind regards,

Igor B. Rogozin

Academic Editor

PLOS ONE

Additional Editor Comments (optional):

The authors implemented suggestions, the paper is acceptable.
---

## [Editor Report · Acceptance letter]

20 May 2020

PONE-D-20-04197R1 

Brain-related genes are specifically enriched with long phase 1 introns 

Dear Dr. Baulin:

I am pleased to inform you that your manuscript has been deemed suitable for publication in PLOS ONE. Congratulations! Your manuscript is now with our production department. 

With kind regards,

on behalf of

Dr. Igor B. Rogozin 

Academic Editor

PLOS ONE